# Whole-Genome Identification of Regulatory Function of CDPK Gene Families in Cold Stress Response for *Prunus mume* and *Prunus mume* var. Tortuosa

**DOI:** 10.3390/plants12132548

**Published:** 2023-07-04

**Authors:** Runtian Miao, Mingyu Li, Zhenying Wen, Juan Meng, Xu Liu, Dongqing Fan, Wenjuan Lv, Tangren Cheng, Qixiang Zhang, Lidan Sun

**Affiliations:** 1Beijing Key Laboratory of Ornamental Plants Germplasm Innovation and Molecular Breeding, National Engineering Research Center for Floriculture, Beijing Laboratory of Urban and Rural Ecological Environment, School of Landscape Architecture, Beijing Forestry University, Beijing 100083, China; 2Center for Computational Biology, College of Biological Sciences and Technology, Beijing Forestry University, Beijing 100083, China

**Keywords:** cold stress, replication events, transcriptional expression profiles, protein-protein interaction network, gene expression

## Abstract

Calcium-dependent protein kinases (CDPK) are known to mediate plant growth and development and respond to various environmental changes. Here, we performed whole-genome identification of CDPK families in cultivated and wild mei (*Prunus mume*). We identified 14 and 17 CDPK genes in *P. mume* and *P. mume* var. Tortuosa genomes, respectively. All 270 CPDK proteins were classified into four clade, displaying frequent homologies between these two genomes and those of other Rosaceae species. Exon/intron structure, motif and synteny blocks were conserved between *P. mume* and *P. mume* var. Tortuosa. The interaction network revealed all PmCDPK and PmvCDPK proteins is interacted with respiratory burst oxidase homologs (RBOHs) and mitogen-activated protein kinase (MAPK). RNA-seq data analysis of cold experiments show that cis-acting elements in the *PmCDPK* genes, especially *PmCDPK14*, are associated with cold hardiness. Our results provide and broad insights into CDPK gene families in mei and their role in modulating cold stress response in plants.

## 1. Introduction

Calcium ion (Ca^2+^) is a universal secondary messenger of signal transduction in many plant physiological processes, especially in plant response to environmental stimuli and plant growth and development [1]. When plants are subjected to various environmental stresses, the changes of transient fluctuations in cytoplasmic Ca^2+^ concentration are perceived by different calcineurin sensors, including calcium-dependent protein kinases (CDPKs) [2], calmodulins (CaMs), CaM-like proteins and calcineurin B-like proteins (CBLs) [3,4,5]. Accordingly, Ca^2+^ signals can be activated and convert to downstream targets. As a vital sensor-transducer molecule, a typical CDPK harbors four domains, consisting of a variable N-terminal domain with myristoylation or palmitoylation sites [6], an inhibitory-junction domain [7], a Ser/Thr protein kinase domain and a calmodulin-like domain constructed by four EF-hands [8]. Therefore, these special functional domains allow CDPKs to have a dual function of a Ca^2+^ sensor and responder that can directly convert upstream Ca^2+^ signals to target genes through phosphorylation [9].

CDPKs are unique to plants and some protists, but not present in animals and fungi [10]. There are 34 genes that were first identified for the CDPK gene family in *Arabidopsis thaliana* (*A. thaliana*) [11], followed by 31 genes in rice (*Oryza sativa*) [12], 20 genes in wheat (*Triticum aestivum* L.) [13], 31 genes in pepper (*Capsicum annuum*) [3], 28 genes in barley (*Hordeum vulgare* L.) [14], 35 genes of CDPKs in maize (*Zea mays*) [15], and 29 genes in tomato (*Solanum lycopersicum*) [5]. Recently, the CDPK gene family has also been reported in many horticultural plants, such as alfalfa (*Medicago sativa*) [16], grape (*Vitis vinifera*) [17], pineapple (*Ananas comosus*) [18], melon (*Cucumis melo* L.) [4], garden strawberry (*Fragaria x ananassa*) [19], and peach (*Prunus persica*) [20]. Some CDPKs members may exhibit different expression tendency and functions during stress response and development stages, but most CDPKs from these plants were seen to have overlapping functions [21,22,23]. With the in-depth study of CDPK genes in different plants, increasing evidence showed that CDPKs exist in various subcellular locations, which means that they mediate many of the abiotic and biotic stress signal transductions [24].

In plants, CDPKs are engaged in hormone response, and metabolic pathway control, stress signaling like cold, salt, or drought stresses [25]. Through abscisic acid (ABA) and Ca^2+^ mediated stomatal regulations, *AtCPK10* interacted with the heat shock protein HSP1 to cope with the drought stress [26]. *OsCPK24*, a cold-responsive kinase in rice, increased proline and glutathione levels to provide cold resistance [27]. By scavenging the accumulation of reactive oxygen species (ROS) and controlling the expression of stress-related genes, *MdCPK1a* overexpression in tobacco plants improved resilience to cold stress [28]. In peach, *PpCDPK7* interacted with respiratory burst oxidase homologs (*PpRBOHA*) on the cell membrane to keep the intracellular ROS balance to lessen chilling damage [20]. In tomato, cold stress significantly increased the CPK27 expression level along with the generation of abscisic acid (ABA) signaling by energizing the production of ROS and nitric oxide (NO) in addition to mitogen-activated protein kinases (MPK1/2) [29]. Conversely, some CDPKs were negative regulators of stress response. The *CPK23* mutant in *Arabidopsis*, showed greatly enhanced tolerance to drought and salt stresses by reducing stomatal apertures [30]. Heterologous overexpression of *ZmCPK1* reduced plants adaption to the cold tolerance [31]. Taken together, in plant response to stressful environments, CDPKs plays a role in both positive and negative regulation.

*Prunus mume,* known as mei, is a woody plant native to China and has early florescence, colorful corollas, and an attractive plant architecture. Among more than 300 varieties of *P. mume* developed in China and Japan, *P. mume* var. tortuosa has particularly a high ornamental value due to its natural curved branches and is the only variety with tortuous branch [32]. Cold temperature has limited the cultivation of mei in the northern China [33]. Moreover, studies have shown that *P. mume* var. tortuosa had weaker cold resistance because of its branches small water retention [34]. However, the molecular mechanisms of cold stress response are still not clear in *P. mume*. In addition, the CDPK genes of wild and cultivated mei have not been identified and analyzed on a genome-wide scale. Little is known about the gene expression pattern of *PmCDPKs* under cold stress. In this study, we identified 14 *PmCDPK* and 17 *PmvCDPK* genes and systematically analyzed gene structures, chromosomal localization, conserved motifs, phylogenetic relationships, and cis-acting elements. The expression of the *PmCDPKs* under low temperature was investigated by public RNA-Seq data and qRT-PCR analysis. Our study deepens the understanding the function of CDPK genes in response to cold stress.

## 2. Results

### 2.1. Identification of CDPK Family Members

A total of 236 CDPK genes were identified in 13 species, including 14 in *P. mume*, 17 in *P. mume* var. Tortuosa, 13 in *P. avium*, 16 in *P.armeniaca*, 16 in *P. persica*, 19 in *P. yedoensis*, 17 in *P. salicina*, 21 in *P. betulifolia*, 15 in *R. chinensis*, 15 in *R. occidentalis*, 25 in *M. domestica*, 17 in *F. vesca*, and 31 in *P. trichocarpa*. Alternative splicing is an important feature of eukaryotic organisms, which was found in *P. trichocarpa* and *P.armeniaca*, with diverse transcripts of the same CDPK genes performed by a, b, and c. All the CDPK genes were named sequentially according to their location on the chromosomes (Table 1 and Appendix A). The lengths of the PmCPDK coding sequences (CDS) ranged from 1584 (PmCDPK1) to 1893 (PmCDPK13) bp, and their predicted numbers of amino acids ranged from 527 (PmCDPK1) to 630 (PmCDPK13) aa, with corresponding molecular weights (MW) varied from 59.44 (PmCDPK1) to 70.38 (PmCDPK13) kDa, respectively. The majority of PmCDPK members had isoelectric points (pI) values below 7.0, except for PmCDPK14 with pI 9.15 values. In *P. mume* var. Tortuosa, the lengths of CDSs varied from 1584 (PmvCDPK1) to 1797, and the longest CDPK protein (PmvCDPK16) encoded 527 amino acids, while the shortest (PmvCDPK1) only encodes 598 amino acids. The MW ranged from 59.44 (PmvCDPK1) to 66.62 (PmvCDPK16) kDa, with pI in the range of 5.17 (PmvCDPK13) to 9.21 (PmvCDPK15). Furthermore, 5 PmCDPK and 7 PmvCDPK proteins with the myristoylation proteins were predicted in their N-terminus, while 10 PmCDPK proteins and 13 PmvCDPK contained palmitoylation sites. Subcellular localization prediction indicated that most CDPK proteins were localized in the cytoplasm or chloroplasts (Table 1).

### 2.2. Phylogenetic Analysis and Classification of CDPK Genes

A phylogenetic tree was built containing 270 full-length protein sequences to compare the CDPKs of *P. mume* with other plants in *Rosaceae* (Figure 1). Based on the previously reported AtCDPKs, all the CPDK proteins were divided into four clades [11]. Clade I had the most CDPK proteins as 87, and clade IV had the least as 25. Furthermore, Clade I contained 10 AtCDPK members, 11 PtCDPK members, 9 MdCDPK members, 7 PbCDPK members, 4 RcCDPK members, 3 RoCDPK members, 6 CDPK members in *F. vesca*, *P. yedoensis and P. armeniaca*, 5 CDPK members in *P. mume*, *P. mume* var. Tortuosa, *P. persica*, *P. salicina* and *P. avium*, respectively. In Clade II, 74 numbers belonging to 14 species, including 4 PmCDPK and PmvCDPK members and 23 CDPK proteins in *Rosaceae*; Clade III comprised 84 proteins with 4 PmCDPK members, 6 *PmvCDPK* members and 66 CDPK proteins in *Rosaceae*. Clade IV comprised 3 AtCDPK and PtCDPK members, 2 CDPK members in *F. vesca*, *M. domestica*, *P. betulifolia*, *P. mume* var. Tortuosa, *P. salicina*, *R. chinensis* and *R. occidentalis*, one CDPK members in *P. armeniaca*, *P. avium*, *P. mume*, *P. persica* and *P. yedoensis,* respectively (Figure 1, Appendix A).

### 2.3. Structural Characterizations and Protein Domain Conservation Analysis of PmCDPK and PmvCDPK Gene Families

The evolutionary analysis further classified the 31 *PmCDPK* and *PmvCDPK* genes into four groups according to conserved domains among the proteins (Figure 2A). MEME software was further used to predict conserved domains of PmCDPK and PmvCDPK proteins. A total of 18 motif were identified and all PmCDPK and PmvCDPK proteins structural domains were conserved and contain motifs 1–4 and 7. PmCDPK13 and PmvCDPK16 (in clade I), PmCDPK5 and PmvCDPK6 (in clade II) specifically contained motifs 14 and 17 in the N-terminal of the proteins. Motif 18 was specifically in the proteins of clade I expect for PmCDPK9, PmvCDPK11, PmCDPK13 and PmvCDPK16. Motif 16 was specifically in the PmCDPK10 and PmvCDPK12 proteins of clade I. Motif 13 was specifically in the proteins of clade III expect for PmCDPK12 (Figure 2B and Appendix A). The exon-introns organization can indicate the evolutionary relationships within *PmCDPK* and *PmvCDPK* families. Phylogenetic tree analysis showed that the *PmCDPK* and *PmvCDPK* genes of each subfamily had similar gene structures. In Clade I, a total of 6 *PmCDPK* and *PmvCDPK* genes contained 7 exons, *PmCDPK8* and *PmvCDPK9* contained one exons each, while *PmCDPK9* and *PmvCDPK16* contained 8 exons each. There were 8 *PmCDPK* and *PmvCDPK* genes with 8 exons in clades II. In Clade III, 3 CDPK genes contained 9 exons, 4 CDPK genes had 8 exons, *PmCDPK1* and *PmvCDPK1* had 7 exon each, while *PmCDPK12* only contained 6 exon. *PmvCDPK10* was identified with 7 exon in clade IV. However, *PmCDPK14* and *PmvCDPK15* (in clade IV) had the largest number of exons as 12 (Figure 2C). The majority of PmCDPK and PmvCDPK genes have comparable numbers of exons and introns, according to these findings, potentially indicating common genetic evolution.

In order to gain insight into the three-dimensional (3D) protein structure of CDPKs in *P. mumu* and *P. mume* var. tortuosa, homology modeling was performed on all PmCDPKs and PmvCDPKs. All protein structures were modeled with 100% confidence using template proteins. 31 proteins were modeled with coverage ranging from 66% (PmvCDPK16) to 98% (PmCDPK4) (Appendix A). The α-helix contributed to 37–52% of the protein structure, whereas the β-strands varied from 7 to 24% (Appendix A). All CDPK proteins in *P. mumu* and *P. mume* var. tortuosa contained catalytic kinase domains, an inhibitory junction domain (JD) and the N-lobe and C-lobe, respectively with each lobe containing two EF hand motifs (Figure 3). By transmembrane structures analysis, all the PmCDPK and PmvCDPK proteins did not have the transmembrane domain except for PmCDPK3 (Appendix A).

### 2.4. Chromosome Localization of PmCDPK and PmvCDPK Gene Families

Chromosomal location with gene density analyses revealed that 14 *PmCDPK* and 17 *PmvCDPK* genes were unevenly distributed across chromosomes (Figure 4). In *P. mume*, *CDPK* genes were distributed on only seven chromosomes. Chr1, Chr4, Chr5 each carried one CDPK gene, Chr2 and Chr6 possessed two genes each, whereas Chr8 and Chr3 possessed three and four genes, respectively. In *P. mume* var. Tortuosa, 17 CDPK genes were distributed on only 8 chromosomes. Chr1, Chr4, Chr7 each carried one CDPK gene, Chr5 and Chr6 possessed two genes each and three genes were mapped on Chr2 and Chr8, respectively. Four CDPKs were located on Chr3.

### 2.5. Gene Duplication and Synteny Analysis PmCDPK and PmvCDPK Gene Families

To examine the expansion patterns of the *CDPK* gene family, we analyzed the duplicated events within the *P. mume* and *P. mume* var. Tortuosa genomes (Figure 5). We identified 4 gene pairs of *CDPKs* derived from segmental gene duplications in *P. mume* (*PmCDPK3/PmCDPK9*, *PmCDPK4/PmCDPK5*, *PmCDPK6/PmCDPK7* and *PmCDPK7/PmCDPK9*) and 3 pairs in *P. mume* var. Tortuosa (*PmvCDPK5/PmvCDPK6*, *PmvCDPK8/PmvCDPK10*, *PmvCDPK11/PmvCDPK13*), but did not find tandem duplication events. Surprisingly, 17 pairs of collinearity relationships between *PmCDPKs* and *PmvCDPKs* were detected. There was a high collinearity relationship among most of the *P. mume* and *P. mume* var. Tortuosa chromosomes, suggesting that most of the *CDPK* genes in both species had a similar origin and evolutionary process. To further explore the specific retention of *PmCDPK* and *PmvCDPK* genes, their collinearity analyses included *AtCDPKs*, *PpCDPKs*, *MdCDPKs* and *RcCDPKs* were performed using the MCScanX. (Figure 6, Appendix A). A synteny of 16 *P. mume* chromosomes with *A. thaliana* while 19 of *P. mume* var. Tortuosa chromosomes with *A. thaliana.* Similarly, 16 pairs of homologous genes between *P. mume* and *P. persica*, while 17 between *P. mume* var. Tortuosa and *P. persica*. A total of 21 and 25 homologous genes pairs were found between *M. domestica* and *P. mume* and *P. mume* var. Tortuosa, respectively. 12 gene pairs were detected in *P. mume* and *R. chinensis*. The result was the same in *P. mume* var. Tortuosa. The collinear complexity of *P. mume* and *P. mume* var. Tortuosa with *M. domestica* was much higher than that other species, indicating that *P. mume* and *P. mume* var. Tortuosa was more closely related to *M. domestica.*

### 2.6. Promoter Analysis of PmCDPK and PmvCDPK Gene Families

Identification of promoter binding sites helps to understand the function of genes. In this study, the key components of the *PmCDPK* and *PmvCDPK* genes include core promoter elements (Figure 7A) and plant-inducible promoter elements including light response elements, stress response elements and hormone response elements. Transcription-associated cis element (CAAT-box and TATA-box) were conserved in all *PmCDPK* and *PmvCDPK* genes. Among them, the types and numbers of light-responsive elements were the largest, for example, Box 4 was present in the promoter regions of each *CDPK* gene, except for *PmCDPK11* and *PmvCDPK13*. Some *PmCDPK* and *PmvCDPK* gene family had unique light-responsive elements, such as GA-motif, AT1-motif, LAMP-element, Gap-box, 3-AF3 binding site, Box II, chs-CMA1a, which were found only in one *CDPK* gene in *P. mume* and *P. mume* var. Tortuosa respectively (Figure 7B). Notably, the *PmCDPK* and *PmvCDPK* promoters contained 11 elements involved in phytohormone response and stress response (Figure 7C,D, Appendix A). Among these five types of hormone-responsive elements, 86% of the *PmCDPKs* and 82% of the *PmvCDPKs* contained MeJA response elements (CGTCA-motif, TGACG-motif), making it the most frequent hormone-responsive element, the number of abscisic response element is second. 93% of the *PmCDPKs* and 94% of the *PmvCDPKs* contained anaerobically inducible elements (AREs). In addition, 36% of the *PmCDPKs* and 29% of the *PmvCDPKs* contained low-temperature responsive binding site (LTR): *PmCDPK1*, *PmvCDPK1* and *PmvCDPK16* had 2 LTRs each, *PmCDPK3/4/12/13* and *PmvCDPK3/4/17* had only one LTR each. All *CDPKs* contained phytohormone response and stress response elements, suggesting that *PmCDPK* and *PmvCDPK* genes containing these elements may be induced to be express by stresses and plant hormones.

### 2.7. Protein-Protein Interaction Network of PmCDPK and PmvCDPK

The network indicated that all the PmCDPKs and PmvCDPKs proteins were associated with three respiratory burst oxidase homolog proteins (RBOHB, RBOHC, RBOHD), which are key producers of reactive oxygen species (ROS) (Figure 8) [35]. In particular, all the PmCDPK and PmvCDPK proteins were interacted with MAPK, a mitogen-activated protein kinase that play a critical role in cold stress response [36]. Some CDPKs (PmCDPK1/7/12, PmvCDPK1/7/14/17) were interacted with xyloglucan endotransglucosylase/hydrolase 9 (XTH9), which cleaves and reconnects xyloglucan molecules [37]. Additionally, PmCDPK8/11 and PmvCDPK9/13 were predicted to interact with ABA insensitive 5 (ABI5) protein, which functions in the core ABA signaling pathway and regulates the expression of stress-responsive genes [38]. PmCDPK2 and PmvCDPK2 interacted with MYB, a transcription factor positively regulate cold tolerance [39].

### 2.8. Expression Pattern of PmCDPKs in Different Underground and Aerial Tissues 

As illustrated in Figure 9A, all the *PmCDPKs* genes were differentially expressed in flower buds, fruits, leaves, roots and stems. Among them, *PmCDPK5/9/12* were expressed in roots, *PmCDPK7* showed higher expression levels in leaves, *PmCDPK3/6* presented relatively higher expression levels in bud, but their expression levels were low in other tissues. All *PmCDPKs* were expressed during the flower bud dormancy period and also at specific stages of development (Figure 9A, Appendix A). Six genes (*PmCDPK1/7/8/10/12/14*) showed the high level of expression from EDI to EDIII, then abruptly decreased in the NF stage. The expressions of six genes (*PmCDPK2/3/4/5/6/9*) were exhibited specifically high level in the NF stage. *PmCDPK11* and *PmCDPK13* showed the highest level of expression in the EDI and EDII, respectively (Figure 9B, Appendix A).

Expression profiles in stems of cold-insensitive cultivar ‘Songchun’ were analyzed under three low temperatures at three test sites: Beijing (BJ), Chifeng (CF), and Gongzhuling (GZL) (Figure 10, Appendix A). The expression of *PmCDPK6* was not detected and *PmCDPK3* expression was low in RNA-seq. At the Beijing site, a total of seven genes (*PmCDPK2/12/14/7/9/4/10*) showed up-regulation in autumn (7.3 °C) and winter (−5.4 °C) but then decreased in spring (3.2 °C). *PmCDPK1* and *PmCDPK8* showed higher expression in spring, the expression of *PmCDPK5*, *PmCDPK11* and *PmCDPK13* was up-regulated in autumn but down-regulated in winter and spring. The results showed that the gene expression trends at CZL and CF sites were relatively consistent with that in BJ. In general, *PmCDPK12/14/7/9/4/10* genes were up-regulated in autumn and winter, then down-regulated in the early-spring in three sites, while *PmCDPK1* and *PmCDPK8* expression showed the opposite trend, the expression levels of the *PmCDPK2/5/11/13* genes showed different trends with the different test sites, indicating that the expression patterns of these genes were different in response to low temperature (Figure 10A). As shown in Figure 10B, for the same season, the expression of most *PmCDPKs* at different test sites was slightly different, among which *PmCDPK3* was highly expressed at Chifeng site in spring, while *PmCDPK9* was highly expressed at GZL site in winter. More than a half of *PmCDPKs* genes showed a relatively low expression in expression in spring, while *PmCDPK1* and *PmCDPK8* was highly expressed at Beijing site in spring.

### 2.9. RT-PCR of PmCDPKs under Cold Treatment

To further investigate the key *P. mume CDPK* genes involved in cold stress, all identified *PmCDPK* genes that showed a higher expression level in 4 °C low temperature treatment, were selected for further RT-qPCR analysis in cold-tolerant cultivar ‘Meirenmei’ and cold-sensitive cultivar ‘Jinsheng’. It is well known that lysis curves can test the specificity of amplification reactions. Therefore, we determined the suitable annealing temperature for the primers based on the shape of the single peak of the solubility curve and the Tm value. Results from gel electrophoresis plots of PCR amplification products showed that each primer showed a single band, indicating good specificity for use in subsequent qRT-PCR experiments (Appendix A). There were the *PmCDPK3*, *PmCDPK6* and *PmCDPK13* that were not detected, a finding that is consistent with the transcriptome data. A total of 12 *PmCDPK* members showed significant expression profiles differed during the cold stress treatment period in the two cultivars. *PmCDPK1/8/11* changed only slightly in the two cultivars, except that the expression level changed significantly at some point in the cold stress. *PmCDPK2* reached its peak at 12 h (‘Jinsheng’) and 24 h (‘Meirenmei’), respectively. The expression level of *PmCDPK5* in ‘Jinsheng’ was higher than that in ‘Meirenmei’, whereas *PmCDPK4* the genes were highly expressed in the ‘Meirenmei’, by contrast, repressed in the ‘Jinsheng’. The expression levels of these genes (*PmCDPK9/10/12/14)* were significantly increased in the early stage of stress and decreased in the late stage of stress in ‘Meirenmei’, but the changes were not significant in ‘Jinsheng’.

## 3. Discussion

A type of serine/threonine-protein kinase known as CDPK, which plays a pivotal role in controlling plant growth and maturation. However, to date, detailed or complete information of CDPKs in mei had not been obtained. Increasing effort has been made on the genomic research of mei, high quality assemblies of the *P. mume* and *P. mume* var. Tortuosa genomes provided powerful genomic information to explore and identify CDPKs for cultivated and wild mei [40,41]. Among the long genetic improvement history of mei, cold is one of the main constraints that seriously affected the transplantation of mei from the south to the north. When plants are exposed to cold, it leads to various changes in physiology and gene expression patterns [42]. CDPKs have been known for many years to participate in Ca^2+^-related signal transduction induced by cold stress, particularly, isoforms were implicated in ABA-mediated signaling [43]. CDPK-dependent changes in ion flux, metabolic changes, or alterations in gene expression [44]. It has been reported that plants overexpressing some CDPK genes of *A.thaliana* [7], rice [27], and other plant species were shown to exhibit enhanced cold tolerance [45]. Some CDPK substrates have been directely linked to abiotic stress response [46], In view of the research progress on CDPK members involved in plant cold stress responses, the identification of CDPK members in cultivated and wild mei has important implications of the selection of cold tolerant mei plants.

On this basis, we identified 14 and 17 *CDPKs* in *P. mume* and *P. mume* var. Tortuosa at the genome level, respectively, which contained the PF00069 and PF13499 domains. Comparative analysis of both species showed that some CDPK genes are present in *P. mume* var. Tortuosa but absent in *P. mume* (Table 1). For example, *P. mume* var. Tortuosa had one *CDPK* gene each on Ch2/5/7, compared with *P. mume.* In addition, this study also identified a total of 205 *CDPK* genes that have been identified in 11 other species, besides the number of *CDPKs* in these plants varied considerably. Such a wide range of occurrence may be due to the high degree of variation at duplicated genome and ploidy level in plants. Phylogenetic analysis of *Rosaceae* plants suggested similarity in CDPK genes. The CDPK proteins of mei were distributed into four groups through protein sequence analyses (Figure 1), which is consistent with reports of *A. thaliana*, rice, maize [47]. Furthermore, *PmCDPK13*/*PmvCDPK16*, *PmCDPK8*/*PmvCDPK9*, *PmCDPK11*/*PmvCDPK13*, *PmCDPK10*/*PmvCDPK12*, *PmCDPK6*/*PmvCDPK8*, *PmCDPK2*/*PmvCDPK2*, *PmCDPK1*/*PmvCDPK1* and *PmCDPK4*/*PmvCDPK15* showed close associations with one another (Figure 1). Most CDPKs have acylation sites, such as N-myristoylation sites and S-palmitoylation sites, which are thought to be key biological processes affecting a variety of cellular functions through modulation of membrane targeting [48]. The variable N-terminal domain of some CDPKs contain N-myristoylation and palmitoylation, five PmCDPKs and seven PmvCDPKs have N-myristoylation, ten PmCDPK and thirteen PmvCDPKs, which have palmitoylation motifs in *P. mume* and *P. mume* var. Tortuosa (Table 1). The N-myristoylation is the binding site of protein and membrane, while palmitoylation plays an important role in membrane binding stability.

Phylogenetic tree analysis showed that CDPK genes of *P. mume* and *P. mume* var. Tortuosa were close to each other in every branch compared to other species, suggesting a high degree of similarity between cultivated and wild mei, as previously observed in sweet potato [49]. The phylogenetic group members of *PmCDPKs* and *PmvCDPKs* had relatively conserved gene structure with only a few exceptions, which are similar to those reports for other gene families in mei, such as GASA [50], SWEET [51], AUXIN/INDOLE ACETIC ACIDs (Aux/IAAs) [52]. In *P. mume* and *P. mume* var. Tortuosa, there are overall differences in the number, length and position of introns, as well as CDS in *CDPK* genes. These variations may result in different lengths of *CDPKs* genes in cultivated and wild mei. There was little variation in the motif composition of CDPKs in both species. Similar protein 3D structural features were obtained in PmCDPK and PmvCDPK (Figure 3), with the typical EF-hand and protein kinase domain motifs (Figure 2), which were also observed in other species, such as *A. thaliana* [11], *Solanum habrochaites* [53], rice, maize, sorghum and ginger (*Zingiber officinale*) [54,55]. This suggests that the CDPK protein structure had conserved function and structural mechanism. The uneven distribution of *CDPKs* on chromosomes is a common feature in *Rosaceae* (Figure 4) and is similar in strawberry [19] and peach [20], which may be related to species evolution and genetic variation. These findings indicate a close relationship and high level of sequence similarity between *P. mume* and *P. mume* var. Tortuosa, suggesting that despite significant differences, they are quite similar at the genomic level, with sequences conserved from wild to cultivated species.

During evolution, duplication of genes is considered to be the main factor in the evolution and diversification of angiosperms. Differences between species become more pronounced especially when species are subjected to selection pressure and restrictive growth conditions [56]. No tandem duplication involving *PmCDPK* and *PmvCDPK* genes was discovered, but 4 segmental duplications and 3 segmental duplications were found in *P. mume* and *P. mume* var. Tortuosa, respectively (Figure 5). Therefore, segmental duplication is the main driving force of duplication for CDPK gene families in two species. It has been reported that segmental duplication as a major factor leading to the expansion of the CDPKs in other species, i.e., grapevine [57], tomato [53]. These duplicated genes may decrease the probability of extinction, as well as contribute to improved stress tolerance in plants [58]. In addition, 18 pairs of collinear genes were detected between *PmCDPK* and *PmvCDPK*, and there was a high collinearity relationship among *P. mume* and *P. mume* var. Tortuosa chromosomes, suggesting that most of the *CDPK* genes in both species had a similar origin and evolutionary process (Figure 5). Compared to that with *A. thaliana* and other *Rosaceae* plants, the collinearity between apple and mei showed higher complexity in CDPK gene family, which is a logical reason for the relatively high number of *MdCDPKs* (Figure 6), indicating that as an ancient tetraploid plant recent whole-genome duplication events and the increase of chromosomes have led to a significant expansion of the apple CDPKs family.

PlantCARE analysis revealed the presence of various cis-acting elements of stress, growth, light, phytohormones and development-related elements in the promoters of *PmCDPK* and *PmvCDPK* genes. These cis-acting elements can help analyze gene function by predicting the binding sites of transcription factors in the promoter regions of genes. In this study, CDPK genes family possesses two key promoter elements including TATA box and CAAT box (Figure 7A). TATA box connects transcription initiation with RNA polymerase and CAAT box regulates gene transcription efficiency. The results indicated that *PmCDPK* and *PmvCDPK* can be transcribed normally. Many CDPKs have been identified in different species in relation to plant cold resistance. A total of 10 *PmCDPK* and *PmvCDPK* genes contained one or more low-temperature responsive cis-elements, suggesting that these genes may be involved in the regulation of low-temperature stress. Moreover, the promoter regions of CDPK members contain different stress-responsive cis-acting elements, suggesting that different *PmCDPK* and *PmvCDPK* memebers may play important roles in different stresses (Figure 7B–D).

Our interaction network predicted that *PmCDPKs* and *PmvCDPKs* were associated with RBOHB, RBOHC and RBOHD. RBOHs are integral membrane proteins that produce superoxide anions (·O^2−^) and subsequently promote the production of ROS, which play a virtual role in the cold response of plants [35]. *FvRBOHD* can be rapidly induced to be expressed in response to cold stress in strawberry [59]. CDPKs have been shown to phosphorylate the N-terminal fragment of RBOHs to participate in signal transmission in defense responses. Other predicted interacting proteins from PmCDPKs and PmvCDPKs network analysis include PbrMAPKs and PtrXTH proteins increased the cold resistance in pears (*Pyrus × bretschneideri*) and poplar (*Populus simonii × Populus nigra)*, respectively [36,37]. 

To further validate the function of CDPK genes, we investigated the expression pattern of CDPK genes under cold stress conditions using RNA-seq data (previously published) and RT-qPCR analysis. In this study, the expression pattern at different tissues in *P. mume* showed tissue-specific expression of *CDPKs* genes, whereas some *PmCDPK* genes (*PmCDPK5/10/12/14*) showed enhanced expression in all tissues (Figure 8A), indicating that they might have a wide range of regulatory functions. Similar tissue-specific expression patterns of *CDPK* genes were also observed in grape and strawberry [57,60]. *PmCDPK3* and *PmCDPK6* were only expressed in the floral buds, not detected in any other tissue, and they may be involved in regulating bud development, dormancy and so on (Figure 9B). The expression levels of the *PmCDPK* gene family members were up-regulated or down-regulated in different seasons and locations (Figure 10), suggesting that *CDPK* gene members play different roles in response to cold stress. Many CDPK genes have been identified to be closely associated with cold stress. In rice, *OsCDPK13* and *OsCDPK24* are positive regulators of cold stress tolerance, whereas *OsCPK17* gene expression reduces cold tolerance but dose not affect the expression of key cold stress-inducible genes [27,61,62]. In grapevine, overexpressing the *VaCPK20* gene increases the expression of stress-responsive genes such as *COR47*, *NHX1*, *KIN1*, or *ABF3* to improve cold resistance [57,63]. In ripe strawberry fruit, *FaCDPK1* transcript levels were increased in response to low temperature [60]. In cucumber and melon [4], almost all *CsCDPKs* and *CmCDPKs* were up-regulated under cold stress [4,64]. In this study, the *PmCDPK14*, which was a homologue of *CDPK28* in *A. thaliana*, was highly expressed in both flower buds and stems at different sites under cold stress. In addition, qRT-PCR analysis showed that *PmCDPK14* gene reached its peak value at 4h of cold tolerance cultivar ‘Meirenmei’, which was 8.06 times that of the control, and reached its peak value at 12 h of cold-sensitive cultivar ‘Jinsheng’, which was 4.7 times that of the control (Figure 11). *AtCDPK28* is activated rapidly upon cold shock and then phosphorylates, leading to promote the nuclear translocation of NIN-LIKE PROTEIN 7 (NLP7), which specifies transcriptional reprogramming of cold-responsive gene sets in response to Ca^2+^ [65]. These results suggest that *PmCDPK14* involved in regulating cold-related genes promotes cold tolerance in ‘Meirenmei’. Previous studies have shown that *PeCPK10* was rapidly induced when the transgenic lines were subjected to −4/−8 °C for 8 h and showed enhanced freezing tolerance [45]. The observed gene expression patterns showed that some *PmCDPKs* (*PmCDPK7/9/10/14*) were more highly expressed in winter than in autumn and spring. Through gene expression patterns analysis, we speculated that these genes played a role in cold stress and freezing stress.

qRT-PCR analysis indicated that three *PmCDPK* genes (*PmCDPK3/6/13*) were considered not to be expressed, which was consistent with the transcription data. *PmCDPK* genes was transcriptionally activated at 4℃ low temperature and increased or decreased in expression with the extension of treatment time. *PmCDPK2/5/7/9/10/12/14* were up-regulated, *PmCDPK1* was negatively regulated under cold stress, suggesting that these genes might be positively or negatively regulated by cold stress. The discrepancy in expression patterns between *PmCDPK4/8/11* is potentially due to genetic differences.

## 4. Materials and Methods

### 4.1. Plants Genome Resources and Identification of CDPK Genes

The reliable genome assemblies of the *P. mume* were obtained from the *Prunus mume* genome database (https://github.com/lileiting/prunusmumegenome, accessed on 5 September 2022). The complete genomes of *Prunus mume* var. Tortuosa (*P. mume* var. Tortuosa), *Prunus persica*, *Prunus avium*, *Malus domestica*, *Fragaria vesca*, *Prunus armeniaca, Rubus occidentalis*, *Prunus salicina*, *Prunus yedoensis*, *Rosa chinensis*, *Pyrus betulifolia*, *Populus trichocarpa (P. trichocarpa)* were downloaded from Genome Database for Rosaceae (GDR) (http://www.rosaceae.org/, accessed on 10 September 2022).

The sequences of 34 AtCDPKs and 31 OsCDPKs were searched from the *Arabidopsis* Information Resource (TAIR, https://www.arabidopsis.org, accessed on 25 September 2022) and the rice genome annotation database (http://rice.uga.edu/, accessed on 28 September 2022). Theses initial sequences was used as query sequences to identify candidate CDPK genes by using the local library software BLAST-P [66] with an E-value of 1 × 10^−7^. Then, all non redundant sequences were verified the sequence by HMMER software (version 3.0, http://hmmer.org/, accessed on 30 September 2022) and the E-value was set up less than 1 × 10^−7^. Finally, NCBI-CDD database (https://www.ncbi.nlm.nih.gov/cdd, accessed on 2 October 2022) and SMART database (http://www.smart.embl-heidelberg.de/, accessed on 10 October 2022) was used to scan and delete the proteins without domains. ExPASy (https://www.expasy.org/, accessed on 15 October 2022) were used to predict the MW, PI and the N-terminal myristoylation sites of *PmCDPKs* and *PmvCDPKs*. GPS-Palm (http://gpspalm.biocuckoo.cn/, accessed on 17 October 2022) and WoLF PSORT (https://wolfpsort.hgc.jp/, accessed on 23 October 2022) was used to predict palmitoylation sites and subcellular localization, respectively [48]. The 3D structures of all PmCDPK and PmvCDPK proteins were predicted by homology modeling using the PHYRE2 web portal (http://www.sbg.bio.ic.ac.uk/phyre2, accessed on 28 October 2022). PHYRE2 uses advanced remote homology detection methods to construct 3D models of protein sequences, with a single highest score template model for all proteins and 100% model confidence.

### 4.2. Phylogenetic and Collinearity Analysis

Alignment of full-length CDPK protein sequences from 14 species MAFFT v7 (https://mafft.cbrc.jp/alignment/server/, accessed on 21 October 2022) [67], Phylogenetic trees were constructed using the maximum likelihood method by MEGA (version 6.0) and 1000 bootstrap repeats were presented. Collinearity analysis was performed using MCscanX software [68] to detect the collinearity pattern of CDPKs among *P. mume*, *P. mume* var. Tortuosa, *A. thaliana*, *P. persica* as well as *M. domestica* and *R. chinensis* [69].

### 4.3. Gene Structure, Conserved Motifs, Chromosomal Location, and Promoter Analyses

MG2C (http://mg2c.iask.in/mg2c_v2.0/, accessed on 1 November 2022) was used to draw a chromosomal location figure. The gene structures of *PmCDPK* and *PmvCDPK* genes were analyzed using the online tool GSDS (http://gsds.cbi.pku.edu.cn/, accessed on 4 November 2022). Conserved motifs of *PmCDPK* and *PmvCDPK* proteins were confirmed using the program MEME. The upstream sequence (2000 bp) of the *PmCDPK* and *PmvCDPK* gene were extracted and promoter elements were predicted using PlantCARE (http://bioinformatics.psb.ugent.be/webtools/plantcare/html/, accessed on 10 November 2022).

### 4.4. Protein Interaction Network of CDPKs

The protein interaction network of PmCDPK and PmvCDPK was predicted by STRING (v 11.5, https://cn.string-db.org/, accessed on 15 November 2022) and was visualized using Cytoscape [70].

### 4.5. Transcriptome Data Analysis

RNA-seq datasets of *PmCDPKs* in five different tissue (flower buds, fruits, leaves, roots, and stems) [40] and flower bud dormancy of *P. mume* (‘Zaolve’) from November to February [71] were downloaded. The expression profiles of *PmCDPK* genes in the stem of *P. mume* cultivar ‘Songchun’ were analyzed in different geographical locations including Beijing (BJ, 39°54′ N, 116°28′ E), Chifeng (CF, 42°17′ N, 118°58′ E) and Gongzhuling (GZL, 43°42′ N, 124°47′ E) and for three different periods of cold acclimation (October, autumn), the final period of endo-dormancy (January, winter), and deacclimation (March, spring) [72].

### 4.6. qRT–PCR Analysis

The annual branches of the cold-tolerated cultivar ‘Meirenmei’ and the cold-sensitive cultivar ‘Jinsheng’ were used as experimental materials and were maintained in water overnight at 22 °C. Plants were treated in a light incubator at 4 °C (16-h light/8-h dark) and sampled for RNA extraction after 1, 3, 6 12, 24, 36, 48, and 72 h.

Total RNA isolation and cDNA synthesis were performed using RNAprep Pure Plant Plus Kit (Tiangen, Beijing, China) and SYBR^®^Green Premix Pro Taq HS qPCR Kit (AccurateBiology, Hunan, China). The reaction system consisted of 20 µL with a 10 µL SYBR^®^Green Premix Pro Taq HS qPCR Kit, 0.4 µL forward and reverse primers mix, 7.2 µL ddH_2_O and 2 µL 10 × cDNA samples. The PCR amplification conditions was as follows: (1) 95 °C for 30 s; (2) 95 °C for 5 s, 55–60 °C for 30 s, 72 °C for 30 s for 40 cycles; (3) 72 °C for 30 s. The relative expression levels of the genes were calculated using the 2^−∆∆Ct^ method [73]. The primers used in this study were listed in Appendix A, the specificity tests of the primers were shown in Appendix A.

## 5. Conclusions

The present research is the first in-depth and methodical report of genome-wide characterization of *CDPK* gene families in cultivated and wild mei. We identified 31 high confidence *CDPK* genes in the genomes of *P. mume* and *P. mume* var. Tortuosa, which were divided into four subgroups based on a phylogenetic analysis. Gene motifs, structure, chromosomal location and cis-acting elements were analyzed to explain the various and traits of the identified CDPK genes. Duplication events occurred in both *P. mume* and *P. mume* var. Tortuosa genomes were identified by collinearity analysis. Importantly, RNA-seq data in five tissues and geographic locations revealed some tissue-specific expression and significant up-regulation of cold responsive CDPK genes. Different expression trends were also observed following RT-qPCR under cold stress, indicating the important role of PmCDPKs in the cold stress signaling pathway. Ultimately, the knowledge gained from this research will contribute to breed cold-tolerance cultivars of mei.

## Figures and Tables

**Figure 1 plants-12-02548-f001:**
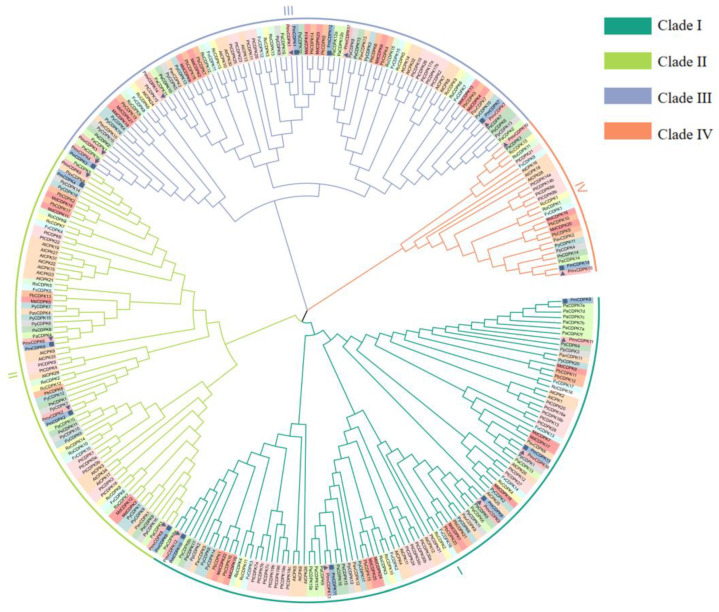
Maximum likelihood phylogeny of the CDPKs family in *A. thaliana* (At), *P. trichocarpa* (Pt), *P. mume* (Pm), *P. mume* var. Tortuosa (Pmv), *P. salicina* (Ps), *P. armeniaca* (Pa), *P. persica* (Pp), *P. avium* (Pav), *P. yedoensis* (Py), *M. domenstica* (Md), *R. chinensis* (Rc), *F. vesca* (Fv)*, P. betulifolia* (Pb), *R. occidentalis* (Ro) was reconstructed.

**Figure 2 plants-12-02548-f002:**
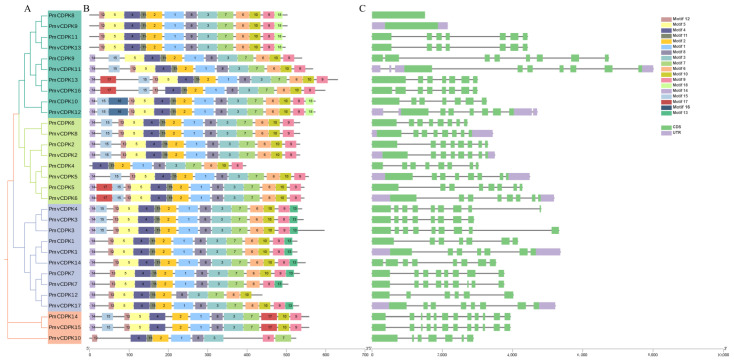
Phylogenetic relationship, motif and structure of PmCDPK and PmvCDPK genes. (**A**) Phylogenetic tree analysis of 14 PmCDPK and 17 PmvCDPK members. (**B**) Schematic diagram of the conserved motifs of PmCDPK and PmvCDPK. Each motif is presented by a particular color. (**C**) Exon-intron distributions of PmCDPK and PmvCDPK genes. Exons are indicated by boxes. Black lines indicate an intron.

**Figure 3 plants-12-02548-f003:**
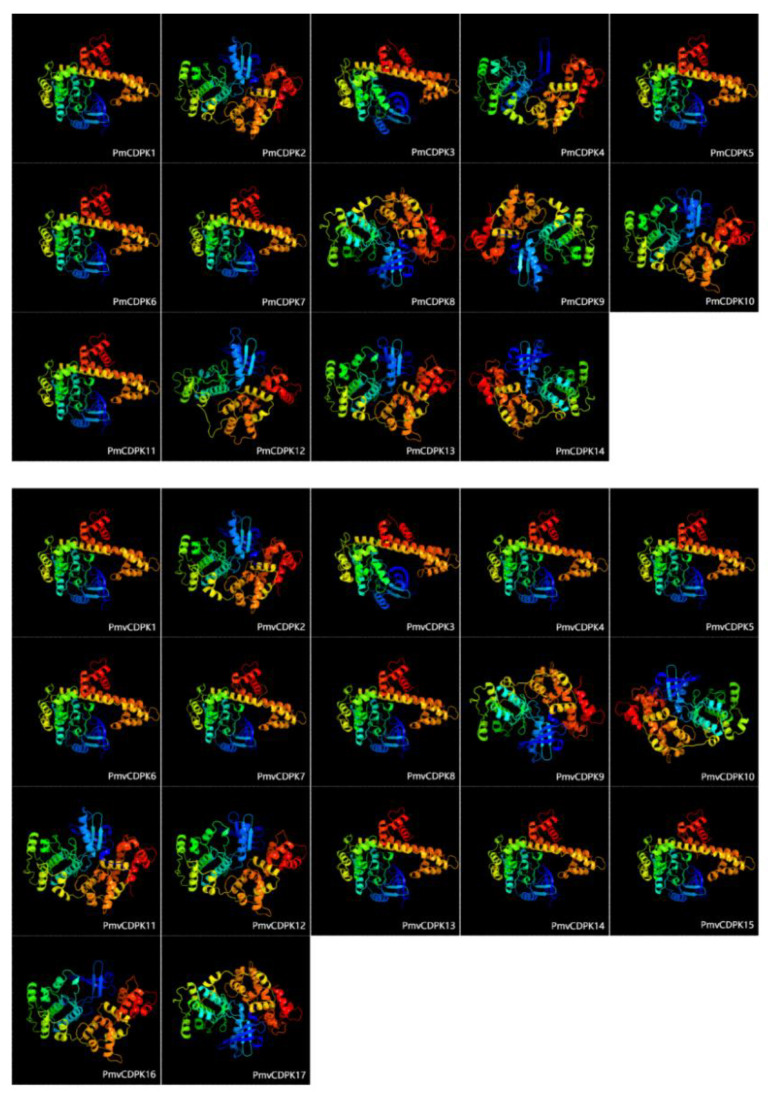
Three-dimensional structure of PmCDPK and PmvCDPK proteins. Each CDPK protein consists of a variable number of α-helix, β-strands, transmembrane helix, and disordered region.

**Figure 4 plants-12-02548-f004:**
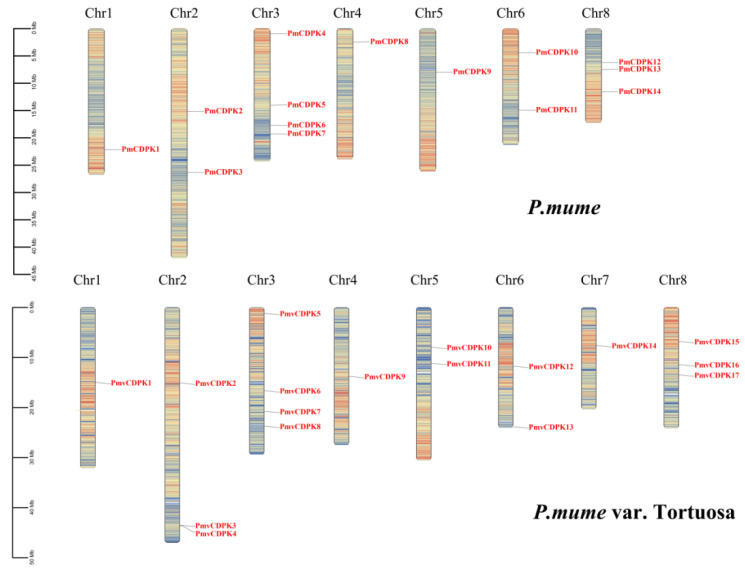
Distribution of *PmCDPKs* and *PmvCDPKs* on chromosomes. The scale bar indicates the length in megabytes (Mb), and the position of each *PmCDPK* (**top**) and *PmvCDPK* (**bottom**) is marked by a black line with the gene ID.

**Figure 5 plants-12-02548-f005:**
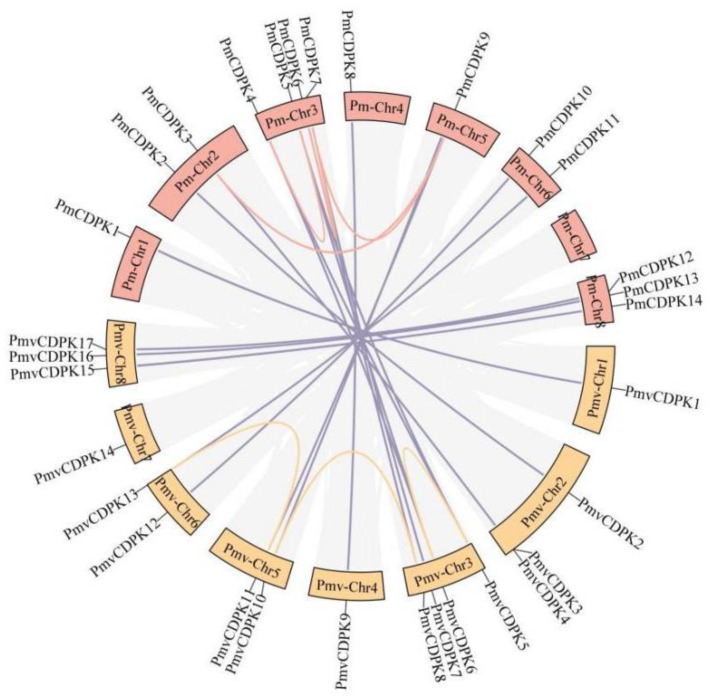
Syntenic relationships among *PmCDPK* and *PmvCDPK* genes. Red and yellow curves mean the synteny relationship of CDPK genes in the *P. mume* and *P. mume* var. Tortuosa genomes, respectively. Violet curves mean the synteny relationship among *PmCDPK* and *PmvCDPK* genes.

**Figure 6 plants-12-02548-f006:**
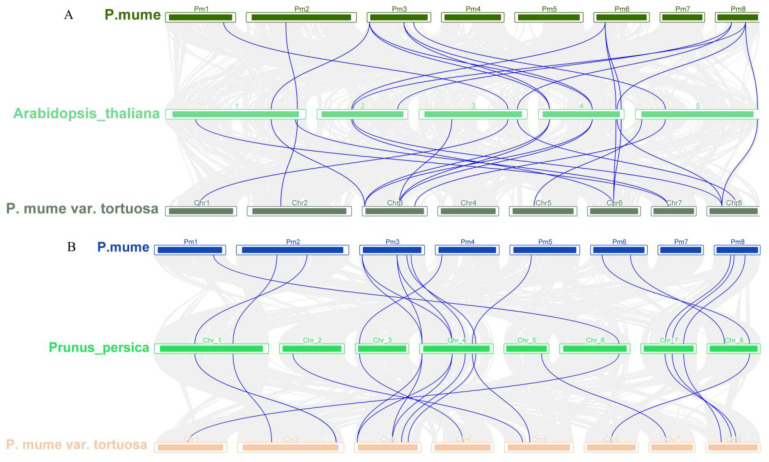
Collinearity analysis of CDPKs in different genomes (**A**) *P. mume* vs. *A. thaliana* vs. *P. mume* var. Tortuosa (**B**) *P. mume* vs. *P. persica* vs. *P. mume* var. Tortuosa (**C**) *P. mume* vs. *M. domestica* vs. *P. mume* var. Tortuosa (**D**) *P. mume* vs. *R. chinensis* vs. *P. mume* var. Tortuosa.

**Figure 7 plants-12-02548-f007:**
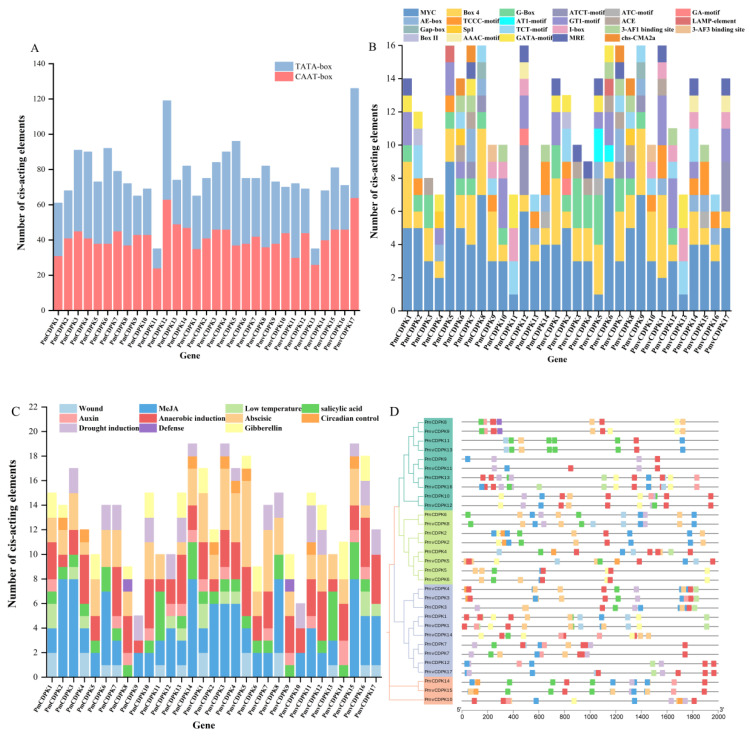
Cis-acting regulatory elements analysis. (**A**) The number of CAAT-box and TATA-box of the *PmCDPK* and *PmvCDPK* genes. (**B**) The number of photoresponsive functional elements in the promoters of the *PmCDPK* and *PmvCDPK* genes. (**C**) The number of stress-related cis elements in the promoters of the *PmCDPK* and *PmvCDPK* genes. (**D**) Phylogenetic tree analysis of 14 PmCDPK and 17 PmvCDPK members (right). The location of stress-related cis elements in the promoters of the *PmCDPK* and *PmvCDPK* genes (left).

**Figure 8 plants-12-02548-f008:**
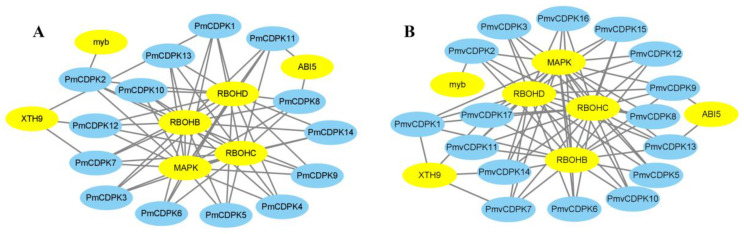
Protein-protein interaction network of PmCDPKs (**A**) and PmvCDPKs (**B**).

**Figure 9 plants-12-02548-f009:**
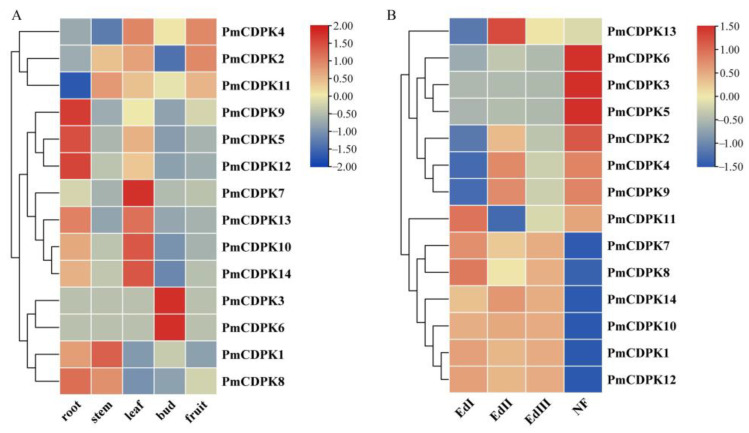
Heat map of RNA-Seq expressions of *PmCDPK* genes. (**A**) Expression profiles of *PmCDPKs* in flower buds, fruits, leaves, roots and stems tissues (**B**) *PmCDPKs* expressions in flower buds of in three endodormancy stages EDI (Endodormancy I, 0% flush rate), EDII (Endodormancy II, 45% flush rate), EDIII (Endodormancy III, 100% flush rate), and NF stage (Natural Flush, flower buds with green tips and dormancy completely released) under low temperature.

**Figure 10 plants-12-02548-f010:**
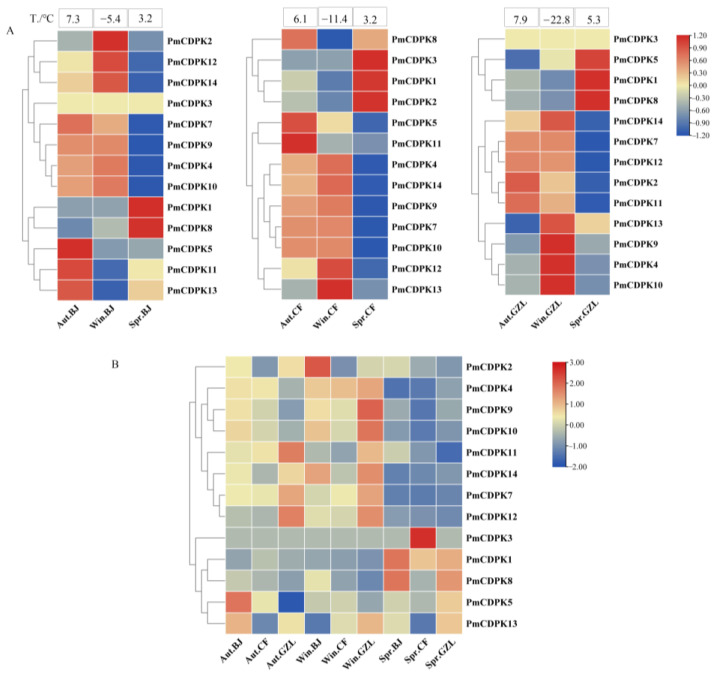
Heat map of RNA-Seq expressions of PmCDPK genes in stems of ‘Songchun’ in different seasons (autumn, winter and spring) and regions (Beijing, Chifeng and Gongzhuling). (**A**) Expression profifiles of PmCDPKs in different seasons at the same test site. (**B**) Expression profifiles of PmCDPKs in different test sites. Aut, Autumn; Win, Winter; Spr, Spring; BJ, Beijing; CF, Chifeng; GZL, Gongzhuling.

**Figure 11 plants-12-02548-f011:**
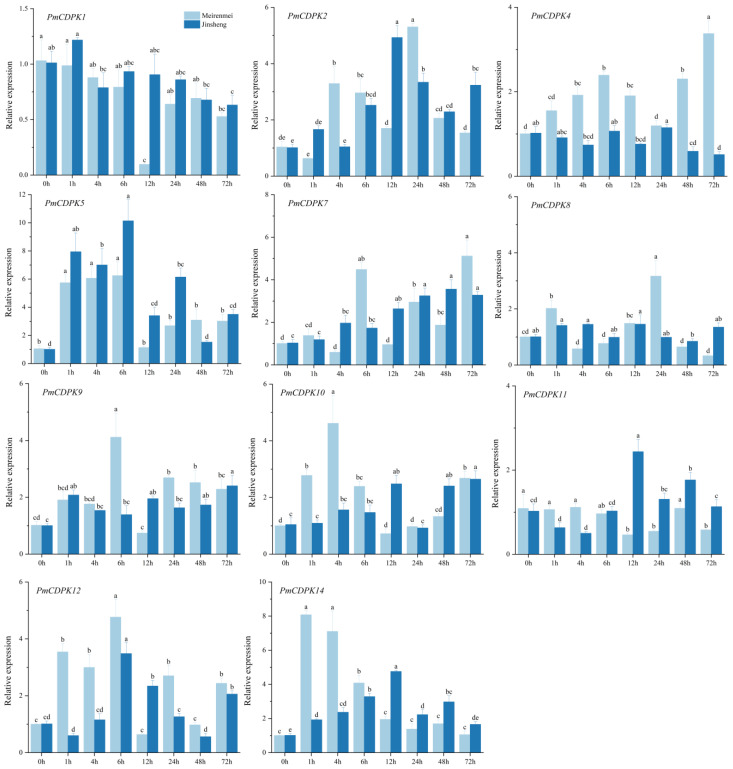
qRT-PCR analysis of *PmCDPKs* in annual branch of cold-tolerant cultivar ‘Meirenmei’ and cold-sensitive cultivar ‘Jinsheng’ under cold treatment. Data depict mean and standard deviation of three replicates (*n* = 3). Different letters denote significance at *p* < 0.05 using Duncan’s multiple range test.

**Table 1 plants-12-02548-t001:** Basic information of CDPK genes in *P. mumu* and *P. mume* var. tortuosa.

Gene Name	Gene ID	Group	CDS Length (bp)	Chromosome Location	Chromosome Starting Position	Chromosome Termination Position	Protein Length (aa)	pI	Molecular Weight (kDa)	N-Myristoylation	Palmitoylation	Subcellula Localization
PmCDPK1	Pm002913	III	1584	1	22,164,004	22,168,153	527	5.95	59.44	N	Y	Cyto.
PmCDPK2	Pm006154	II	1605	2	15,149,171	15,152,466	534	6.07	60.41	Y	Y	Chlo.
PmCDPK3	Pm007589	III	1791	2	26,332,496	26,337,814	596	5.49	67.49	Y	Y	Plas.
PmCDPK4	Pm009790	II	1194	3	940,220	943,249	397	5.14	44.85	ND	ND	E.R.
PmCDPK5	Pm011831	II	1638	3	14,012,305	14,016,582	545	6.38	61.19	Y	Y	Nucl.
PmCDPK6	Pm012231	II	1605	3	17,715,387	17,718,099	534	5.72	59.76	Y	Y	Nucl.
PmCDPK7	Pm012379	III	1602	3	19,277,893	19,281,650	533	5.95	59.92	N	Y	Cyto.
PmCDPK8	Pm013203	I	1509	4	2,459,927	2,461,435	502	5.14	56.51	ND	ND	Cyto.
PmCDPK9	Pm017091	I	1620	5	7,971,082	7,977,817	539	5.23	60.19	N	N	Chlo.
PmCDPK10	Pm020686	I	1722	6	4,404,203	4,407,462	573	5.88	64.24	N	Y	Nucl.
PmCDPK11	Pm022189	I	1494	6	14,941,879	14,946,305	497	5.17	55.77	ND	ND	Chlo.
PmCDPK12	Pm025893	III	1317	8	6,191,749	6,195,770	438	6.34	49.38	N	Y	Cyto.
PmCDPK13	Pm026064	I	1893	8	7,481,694	7,484,694	630	5.94	70.37	N	Y	Chlo.
PmCDPK14	Pm026757	IV	1674	8	11,588,959	11,592,896	557	9.15	62.91	Y	Y	Chlo.
PmvCDPK1	PmuVar_Chr1_1688	III	1584	1	15,005,082	15,009,232	527	5.95	59.44	N	Y	Cyto.
PmvCDPK2	PmuVar_Chr2_2118	II	1605	2	15,091,603	15,094,609	534	6.07	60.39	Y	Y	Chlo.
PmvCDPK3	PmuVar_Chr2_5629	III	1632	2	43,536,639	43,539,536	543	5.58	61.67	Y	Y	Cyto.
PmvCDPK4	PmuVar_Chr2_5639	III	1620	2	43,605,806	43,610,614	539	5.67	61.23	Y	Y	Cyto.
PmvCDPK5	PmuVar_Chr3_0208	II	1671	3	1,179,360	1,182,876	556	6.24	62.26	Y	Y	Chlo.
PmvCDPK6	PmuVar_Chr3_2251	II	1638	3	16,612,214	16,616,505	545	6.38	61.19	Y	Y	Nucl.
PmvCDPK7	PmuVar_Chr3_2694	III	1515	3	20,764,473	20,768,229	504	5.80	56.54	N	Y	Cyto.
PmvCDPK8	PmuVar_Chr3_2937	II	1605	3	23,712,812	23,715,522	534	5.72	59.76	Y	Y	Nucl.
PmvCDPK9	PmuVar_Chr4_1515	I	1509	4	13,779,495	13,781,003	502	5.21	56.50	ND	ND	Chlo.
PmvCDPK10	PmuVar_Chr5_0743	IV	1575	5	7,977,018	7,979,900	524	5.58	58.21	N	N	Cyto.
PmvCDPK11	PmuVar_Chr5_1007	I	1704	5	11,205,040	11,211,809	567	5.34	63.22	N	N	Chlo.
PmvCDPK12	PmuVar_Chr6_1581	I	1722	6	11,753,785	11,756,996	573	5.88	64.24	N	Y	Nucl.
PmvCDPK13	PmuVar_Chr6_3038	I	1494	6	23,843,556	23,847,980	497	5.17	55.77	N	N	Chlo.
PmvCDPK14	PmuVar_Chr7_1057	III	1647	7	7,607,280	7,610,805	548	6.71	62.18	N	Y	Cyto.
PmvCDPK15	PmuVar_Chr8_1181	IV	1674	8	6,815,631	6,819,563	557	9.21	62.90	Y	Y	Chlo.
PmvCDPK16	PmuVar_Chr8_1868	I	1797	8	11,437,704	11,440,698	598	5.75	66.62	N	Y	Chlo.
PmvCDPK17	PmuVar_Chr8_2100	III	1596	8	13,491,294	13,495,563	531	6.25	60.12	N	Y	Cyto.

Note: Subcellular localization: Cyto. Ctyoplasm, Chlo. Chloroplast, Plas. Plasma membrane, E.R. Extracell, Nucl. Nucleus.

## Data Availability

The data are included in Appendix A.

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
