# Peer review of "Whole-Genome Identification of Regulatory Function of CDPK Gene Families in Cold Stress Response for Prunus mume and Prunus mume var. Tortuosa"

_plants, 2023, doi:10.3390/plants12132548_

Round 1
Reviewer 1 Report
the framework of the study is very interesting. the introduction is well organized and gives enough information about studies with similar research interest. the authors made effort to establish the correlation between specific genomes and plants responses to cold stress. the figures are very explanatory which makes reader easier to understand and made its own conclusion. however, there is few points for improvements firstly in the methodology in the line 472 and 473 the authors said:" The 3D structures of all PmCDPK and PmvCDPK proteins were predicted by homology modeling using the PHYRE2 web portal (http://www.sbg.bio.ic.ac.uk/phyre2)." but they didn't give accuracy of that prediction model.
Reviewer 2 Report
Dear authors, I, after hesitation, chose to suggest major revision of the paper rather than reject it. I reason that you did quite thorough analysis of the plant CDPK family. There are problems with it, though.
You say you found the CDPK genes in the genomes. In fact, you found some genes that look like CDPK genes. Without demonstration their functions it is not possible to say more than that. Such comments apply to other places in the paper, protein structures for instance. This disclaimer lowers, more exactly, puts in its proper place, the value of the paper’s results, and I believe such changes will restructure the paper a lot. Justify criteria why you call a gene a CDPK gene.
Secondly, you are comparing two close plum varieties, and then you compare this plum with the rest of the plant kingdom. Two too different jobs to fit in one paper. Remove one plum variety I suggest. Save it for another paper.
You do qRT-PCR. Negative (and positive) controls are not found by me. Why do you think you amplify the right gene exclusively? Most papers who do this do address PCR specificity experimentally.
Some of minor problems are noted below.
line 110 genes or gene products?
line 116 clade or subgroup? Use the term consistently
line 178 non-random Why?
line 265 Fig 8A or 9A?
I look forward to revised version.
Reviewer 3 Report
The manuscript “plants-2300958” entitled “Whole-Genome Identification of Regulatory Function of CDPK Gene Families in Cold Stress Response for P. mume and P. mume var. Tortuosa” by Miao et al. provides an interesting study where it was performed whole-genome identification of CDPK families in cultivated and wild mei, Prunus mume. The study identified 14 and 17 CDPK genes in P. mume and P. mume var. Tortuosa genomes, respectively. All 270 CPDK proteins were classified into four clades, displaying frequent homologies between these two genomes and those of other Rosaceae species. Exon/intron structure, motif, and synteny blocks were conserved between P. mume and P. mume var. Tortuosa. RNA-seq data analysis of cold experiments showed that cis-acting elements in the PmCDPK genes, especially PmCDPK14, are associated with cold hardiness. The results of the present study provide broad insights into CDPK gene families in mei and their role in modulating cold stress response in plants.
For publication in the “Plants” journal, the topic and content are appropriate. The subject of the study is interesting and topical, with high scientific and practical importance. The introduction is in accordance with the subject and correctly presented. Numerous scientific articles of recent date and in concordance with the topic of the study were consulted. The methodology of the study was clearly presented, and appropriate to the proposed objectives. The obtained results have been fully analyzed. The scientific literature, to which the reporting was made, is recent and representative in the field. The editing and linguistic quality are good. In addition, it is easy to follow by the reader, the figures and tables give good summaries, and the text editing to a thoughtful conclusion part. However, However, the review of the article revealed some issues, which were noted in the article and listed below:
· Title and abstract: When referring to species for the first time, please write the full name of the species (Prunus mume instead of P. mume) as in the last paragraph of the introduction.
· Keywords: Please change some keywords. The title and keywords must not contain the same words.
· Lines 46: Scientific name of wheat must be italicized.
· Figures 1, 2, 6, and 7: The quality of the figures is low and blurred. Please fix this problem.
· Discussion: The discussion section must be enhanced. The authors should further refer to previous studies concerning the function of the CDPK family in other plant species.
· Finally, the reviewer recommends the authors carefully revise the paper format and back matter section (author contributions, funding, etc.) and be consistent with the formatting of references and cross-references. This has to be standardized across the paper. For more details, please see “Instructions for authors”
Thank you for your consideration.
Reviewer 4 Report
Excellent work
Author Response
Thank you for your decision on our manuscript entitled “Whole-Genome Identification of Regulatory Function of CDPK Gene Families in Cold Stress Response for Prunus mume and Prunus mume var. Tortuosa”. We have carefully considered the suggestion of Reviewer and make some changes. We have tried our best to improve and made some changes in the manuscript.
Thank you very much for your consideration.
I look forward to hearing from you.
Yours sincerely,
Lidan Sun, PhD
Professor of Forest Genetics
Beijing Forestry University
Round 2
Reviewer 2 Report
Dear authors,
You say "we screened the specificity of primers and set a series of annealing temperatures to determine the most appropriate annealing temperature and to ensure the specificity of the primers". I need to see it in the paper or in the supplement. Do show that the primers amplify CDPKs. Of course when you raise Tm, the amplification will not go at some point. It is up to you how to prove the specificity, the right size band on a gel for instance.
Round 3
Reviewer 2 Report
I think Figs D and E from your response to me should be in Supplementary materials and referenced in the main text.
Author Response
Thank you very much for your suggestions.
We've included the figures in the additional material and cited it in the article, as shown in lines 550-551 (page 18), and marked in yellow.
Round 4
Reviewer 2 Report
Please give S3 as the name of your figure with the gel
Author Response
Thank you for your suggestion. We have already made modifications according to your requirements. Thanks again!